# On the Challenges of Acoustic Energy Mapping Using a WASN: Synchronization and Audio Capture

**DOI:** 10.3390/s23104645

**Published:** 2023-05-10

**Authors:** Emiliano Ehecatl García-Unzueta, Paul Erick Mendez-Monroy, Caleb Rascon

**Affiliations:** 1Posgrado de Ingeneria Electrica, Universidad Nacional Autónoma de México (UNAM), Universidad 3000, Mexico City 04510, Mexico; 2Unidad Académica del IIMAS en el Estado de Yucatán, Universidad Nacional Autónoma de México (UNAM), Parque Científico y Tecnológico de Yucatán, Yucatán 97302, Mexico; erick.mendez@iimas.unam.mx; 3Instituto de Investigaciones en Matemáticas Aplicadas y en Sistemas (IIMAS), Universidad Nacional Autónoma de México (UNAM), Circuito Escolar 3000, Mexico City 04510, Mexico; caleb@unam.mx

**Keywords:** wireless acoustic sensor network, synchronization, beamforming, acoustic mapping

## Abstract

Acoustic energy mapping provides the functionality to obtain characteristics of acoustic sources, as: presence, localization, type and trajectory of sound sources. Several beamforming-based techniques can be used for this purpose. However, they rely on the difference of arrival times of the signal at each capture node (or microphone), so it is of major importance to have synchronized multi-channel recordings. A Wireless Acoustic Sensor Network (WASN) can be very practical to install when used for mapping the acoustic energy of a given acoustic environment. However, they are known for having low synchronization between the recordings from each node. The objective of this paper is to characterize the impact of current popular synchronization methodologies as part of the WASN to capture reliable data to be used for acoustic energy mapping. The two evaluated synchronization protocols are: Network Time Protocol (NTP) y Precision Time Protocol (PTP). Additionally, three different audio capture methodologies were proposed for the WASN to capture the acoustic signal: two of them, recording the data locally and one sending the data through a local wireless network. As a real-life evaluation scenario, a WASN was built using nodes conformed by a Raspberry Pi 4B+ with a single MEMS microphone. Experimental results demonstrate that the most reliable methodology is using the PTP synchronization protocol and audio recording locally.

## 1. Introduction

The analysis of the acoustic scene can be used in multiple applications: urban monitoring [1], where acoustic sensors capture information from the environment and compares it with known events to determine urban activity; bio-localization [2], where microphones arrays are deployed to determine the position of sound emitting animals; rescue work in catastrophe situations [3,4], where the sensors are deployed to measure acoustic activity of to-be-rescued entities to determine their position; domestic smart systems [5], where sensors are used to locate the user that is to be serviced inside a domestic environment; etc. In such scenarios, the acoustic information of the environment (sources of interest, noise, interference and reverberation) is obtained by processing acoustic signals captured by a monitoring system [6,7] as shown in Figure 1. The problem of detecting of acoustic events for the mentioned applications has been approached by implementing a network of sensors that capture information from the environment and processing it to determine the type of acoustic event and its position.

With this configuration, different mathematical models can be used to generate what is known as an ’acoustic energy map’, which provides information that could be used to localize sound sources, to establish how ’loud’ they are, etc. To carry this out, an acoustic sensor network is usually used to sample the acoustic scene [8]. However, approaching the problem like this requires a vast amount of nodes (or microphones) in the acoustic network. Thus, we propose to employ mathematical models that rely on the concept of beamforming which do not require as many nodes in the acoustic network to provide an acoustic energy map, but heavily depend on the difference in time of arrival of an acoustic source to each of the nodes of the acoustic network. These acoustic mapping model is described in Figure 2, where $x_{i}\left(t\right)$ is the acoustic signal captured for the *i*th node of the network, and an acoustic source is assumed to exist in a point (x,y). The signal is formed by applying a beamformer with which it is possible to determine the energy in that point. Then, a grid of candidate points (or proof points) are proposed and for each one its energy is calculated and plotted to obtain the acoustic energy map.

Additionally, it is of interest to employ a Wireless Acoustic Sensor Network (WASN) to capture the multi-channel data from which the acoustic energy map is estimated, because of their ease of installation and modification due to its nodes not being tethered by any physical cabling. However, the recordings may be affected by synchronization issues which, in turn, may affect the performance of the aforementioned beamforming-based acoustic energy mapping technique.

The main goal of the paper is to characterize how different synchronization/recording methodologies impact beamforming-based acoustic energy mapping. The methodologies employed are:Local recording with Network Time Protocol (NTP) synchronization. The WASN is synchronized with NTP and each node records its captured acoustic data locally.Local recording with Precision Time Protocol (PTP) synchronization. The WASN is synchronized with PTP and each node records its captured acoustic data locally.Remote recording with PTP synchronization. The WASN is synchronized with PTP and each node transmits its captured acoustic data to a centralized server.

Then, the shape of the acoustic energy map, as well as the statistical values on the synchronization error in the node timestamps, are employed to characterize these methodologies.

In summary, the overall process presented and characterized in this work of generating an acoustic energy map using a WASN consists on:The design of the WASN: obtained by analyzing the characteristics of different processing units and microphones and selecting the ones that suited the most for the desired application.The implementation of the synchronization protocols: it consisted on the selection of available synchronization protocols for a WASN (NTP and PTP), and their characterization in the creation of an acoustic energy map.The proposal of several acoustic signal recording methodologies: it took in consideration the advantages and disadvantages that each methodology presented and their evaluation in a controlled experiment.The generation of an acoustic energy map through beamforming techniques: there was a stage of research for beamformers techniques and the characterization of the selected ones in the application of acoustic energy mapping.

The paper is structured as follows: Section 2 describes the methodologies used for synchronization; Section 3 describes the acoustic energy mapping methodologies; Section 4 presents the results of several beamforming-based acoustic energy mapping techniques using signals from a simulated WASN and simulated sound sources, artificially inserting synchronization errors; Section 5 presents the results using signals recorded with a real-life WASN, while employing the aforementioned synchronization methodologies; Section 6 discusses the advantages, disadvantages and what are the feasible applications for each synchronization methodology; finally Section 7 presents some concluding remarks.

## 2. Wireless Acoustic Sensor Network and Synchronization

A Wireless Acoustic Sensor Network (WASN) consists of a set of sensing nodes distributed on a physical space that captures data from the acoustic environment. Each node consists of a processing device and an acoustic sensor (microphone) [9,10,11,12]. All nodes are connected through a local wireless network to receive and send data, with the final actionable device being a multi-channel audio recording WASN(each channel being providing by a node in the WASN). The basic design of a WASN with *M* nodes is presented on Figure 3.

As mentioned in the introduction, beamforming-based techniques [13] are meant to be applied towards the generation of an acoustic energy map. Beamformers use the difference in arrival times of the wavefront of the acoustic source to each node to reconstruct the signal at a given position.

Thus, the multi-channel audio recording must be synchronized for them to work properly, since errors in synchronization will affect their performance, the generated acoustic energy map will not represent the actual acoustic environment [14,15].

### 2.1. WASN Design

The WASN that will be used to capture signals to generate the acoustic energy map consisted of a set of wirelessly-interconnected capture nodes, each composed by a processing unit connected to an acoustic sensor (e.g., microphone). Based on previous experimentation with different capture node configurations, a Raspberry Pi 4B+ with a MEMS (micro electret microphone) Microphone Breakout Board was chosen as the processing unit and acoustic sensor, respectively. The processing unit was selected over other models of Raspberry and Arduinos because its features allow the network to capture, send and receive data, while bearing a relatively low cost. The MEMS microphone was selected because of its low cost, its ease of installation and its high sensibility and SNR. The characteristics of both are presented in Table 1 and Table 2.

### 2.2. Synchronization Protocols

Each node in the WASN has its own processing device with its own internal clock which can be affected by physical interactions with the environment, such as: temperature, pressure, vibration, etc. When the acoustic signal is captured, the timestamps of each audio file have to be synchronized. The synchronization represents a problem that can be solved by using extra hardware (like an external clock) or by applying synchronization protocols to synchronize all of the node’s internal clocks [16,17]. There are different strategies that can be implemented to achieve this, most of which use a server to exchange messages with their respective timestamps, which then are used to adjust the internal clocks of the nodes of the WASN. The most commonly used are: Network Time Protocol (NTP) (Figure 4) and Precission Time Protocol (PTP) (Figure 5) [18,19]. Both protocols exchange messages through a network between a master-server and a client-slave. The client-slave sends a request to the server-master and when it responds, the client-slave adjusts its internal clock. The difference between NTP and PTP is that for the latter, there is an extra stage of verification for the synchronization, which results in higher precision.

A synchronization implementation is proposed as follows: one of the nodes of the WASN is the time server for the network, the internal clock of which is the one that all the other nodes will adjust their time to. There is also the possibility for the time server node to synchronize its internal clock with a publicly-available time server.

Although only the the NTP and PTP protocols were implemented, it is left for future work to explore other strategies for synchronization and protocols.

### 2.3. Recording Methodologies

#### 2.3.1. Local Recording Methodology

In this methodology, each node is responsible for capturing acoustic data from the environment, store it, to later send such data to a central processing unit to generate the acoustic energy map. Each node generates its own timestamp according to its internal clock. This methodology is presented in Figure 6.

Different synchronization protocols can be used with this recording methodology, thus, we propose the following:Local recording with NTP. Each client node synchronizes its internal clock with the server node on a background process, and the server node synchronizes its internal clock with a time server service on the internet.Local recording with PTP. It operates on a similar fashion to the previous methodology, but the slave nodes synchronize their internal clocks with the clock from the master.

In this type of recording methodology, each node is responsible for monitoring the acoustic environment and generating and storing audio and timestamps files locally. The data processing for this type of methodology requires the adjustment of the timestamps of each node to a general frame of time. After the adjustment, the beamforming process is carried out to generate the acoustic energy map.

#### 2.3.2. Remote Recording Methodology

This methodology generates only one timestamp for all the audio signals captured by the nodes in the WASN, shown in Figure 7. Hence, instead of storing the recorded data locally, the data is sent through the local network to a central processing unit for storage. The advantage of this methodology is that there is no need for post-processing multiple timestamps. Each node of the WASN only needs to capture the acoustic data and transmit it, reducing the load of all the processing units.

## 3. Beamforming-Based Acoustic Energy Mapping

A beamformer usually employs a series of weights (or steering vector) to estimate the radiation pattern related to a given steered position. These weights are calculated based on the respective time difference of arrival, for the given steered position, for each node in the WANS. To this effect, a collection of points (x,y) is proposed, referred here as “proof points”. A beamformer and its respective set of weights are applied as if there was an acoustic energy source at each proof point. The signal associated to that point is reconstructed, and its energy calculated and recorded for its respective proof point. The result is an acoustic energy map which was “sampled” at each proof point.

The aforementioned time differences depend on the geometry of the WASN and the distance between the acoustic source and each node, as exemplified in Figure 8.

Let the amount of nodes inside the WASN be *M*, thus, the Cartesian coordinates of each can be written as $\left(x_{1},y_{1}\right),\left(x_{2},y_{2}\right),\left(x_{3},y_{3}\right),\cdots ,\left(x_{M},y_{M}\right)$. Let the frequency-domain signal captured at each node *i*th node be $X_{i}$. Additionally, let the acoustic source position be $\left(x_{s},y_{s}\right)$ (which in this case, is the location of a given proof point). The time of arrival of the wavefront of the acoustic source to each *i*th node ($t_{i}$) can be calculated by:(1)$$t_{i}=\frac{\sqrt{{\left(x_{s}-x_{i}\right)}^{2}+{\left(y_{s}-y_{i}\right)}^{2}}}{c}$$
where $c=343\frac{m}{s}$ is the speed of sound in the air. Consequently, the vector of arrival times ($T=[t_{1},t_{2},\cdots ,t_{M}]$) can be expressed as:(2)$$T=\left[\frac{\sqrt{{\left(x_{s}-x_{1}\right)}^{2}+{\left(y_{s}-y_{1}\right)}^{2}}}{c}\frac{\sqrt{{\left(x_{s}-x_{2}\right)}^{2}+{\left(y_{s}-y_{2}\right)}^{2}}}{c}\cdots \frac{\sqrt{{\left(x_{s}-x_{M}\right)}^{2}+{\left(y_{s}-y_{M}\right)}^{2}}}{c}\right]$$

To reconstruct a signal S(x,y) at a given proof point located at (x,y), a series of weights $w_{i}\left(x,y\right)$ are applied to the input signals $X_{i}$, namely:(3)$$S\left(x,y\right)=\sum_{i=1}^{M}w_{i}{\left(x,y\right)}^{H}X_{i}$$
where $w_{i}{\left(x,y\right)}^{H}$ is the complex conjugate factor. As mentioned earlier, the weights are based on the time differences of arrival at each node, for each possible proof point. Their calculation differ between beamforming methodologies. The beamformers proposed for the generation of the acoustic energy map are:Delay and Sum (DAS).Minimum Variance Distortionless Response (MVDR).Steered-Response Power with Phase Transform (SRP-PHAT).Phase-based Binary Masking (PBM).

Once the signal is reconstructed for a particular proof point, its energy is proportional to the signal amplitude squared.
(4)$$E\left(x,y\right)\propto {∥S\left(x,y\right)∥}^{2}$$

The acoustic energy map is then obtained by proposing a grid of proof points and plotting the total energy associated of the signal in a period of time to each of them.

In the rest of these sections, the beamformer techniques are briefly explained, noticing their differences, advantages and disadvantages.

### 3.1. Delay and Sum (DAS)

This is the simplest version of the beamforming techniques, since it relies in shifting the input signals such that the information that is received from a given steered position is aligned across all the input signals [20]. Thus, if summed, the signal in the given steered position is amplified. To this effect, the weights for this beamformer are simply a series of delay operators that counter the occurring time differences of arrival. That is to say, they are calculated as:(5)$$w_{i}\left(x,y\right)=\left[\begin{array}{cccc} e^{-i2\pi \omega_{1}t_{1}} & e^{-i2\pi \omega_{2}t_{1}} & \cdots & e^{-i2\pi \omega_{N}t_{1}}\\ e^{-i2\pi \omega_{1}t_{2}} & e^{-i2\pi \omega_{2}t_{2}} & \cdots & e^{-i2\pi \omega_{N}t_{2}}\\ \vdots & \vdots & \ddots & \vdots \\ e^{-i2\pi \omega_{1}t_{M}} & e^{-i2\pi \omega_{2}t_{M}} & \cdots & e^{-i2\pi \omega_{N}t_{M}} \end{array}\right]$$
where $\omega_{l}$ is the *l*th frequency bin. Usually, $t_{1}$ is set to 0, and all the other $t_{i}$’s are set relative to that time, such that $t_{i}=t_{1}-t_{i}$. It is important to note that this set of weights is known as the `steering vector’, and is re-used in other beamformer techniques.

### 3.2. Minimum Variance Distortionless Response (MVDR)

This technique approaches the problem of reconstructing the signal by aiming to optimize the weights such that the energy that is not coming from a steered position is minimized [21]. It does so by first expressing the output energy (${\left|S\left(x,y\right)\right|}^{2}$) in terms of the co-variance ($R=XX^{H}$) between the input signals ($X=[X_{1},X_{2},\cdots ,X_{M}]$). Due to Equation (Equation 3), and $W\left(x,y\right)=[w_{1}\left(x,y\right),w_{2}\left(x,y\right),\cdots ,w_{M}\left(x,y\right)]$, it is derived that:(6)$${\left|S\left(x,y\right)\right|}^{2}={\left|W{\left(x,y\right)}^{H}X\right|}^{2}=\left(W{\left(x,y\right)}^{H}X\right)\left(X^{H}W\left(x,y\right)\right)=W{\left(x,y\right)}^{H}RW\left(x,y\right)$$

It carries out a minimization of such energy to obtain its optimized weights $W_{op}\left(x,y\right)$, with the following restriction:(7)$$W_{das}{\left(x,y\right)}^{H}W_{op}\left(x,y\right)=1$$
where $W_{das}$ is the steering vector as calculated in Equation (Equation 5). This minimization has the following generalized solution:(8)$$W_{op}\left(x,y\right)=\frac{R^{-1}W_{das}\left(x,y\right)}{W_{das}{\left(x,y\right)}^{H}R^{-1}W_{das}\left(x,y\right)}$$

### 3.3. Steered-Response Power with Phase Transform (SRP-PHAT)

This technique tries to maximize the output signal of a DAS beamformer but in a determined direction, and directly provides the power of the signal that arrives from such direction [22,23].

The power of the output signal S(x,y), as described in Equation (Equation 3), is given by:(9)$$P\left(x,y\right)=\int_{-\infty }^{\infty }{\left|S\left(x,y,\omega \right)\right|}^{2}d\omega$$
where S(x,y,ω) is the ω frequency of S(x,y).

This integral is called the steered-response power and its discrete form can be written as:(10)$$P\left(x,y\right)=\sum_{n=0}^{M-1}\sum_{m=n+1}^{M-1}R_{n,m}\left(\tau_{n,m}\left(x_{s},y_{s}\right)\right)$$
where $\tau_{n,m}$ is the time difference of arrival between nodes *n* and *m* of source located at $\left(x_{s},y_{s}\right)$, and can be calculated similarly as in Equation (Equation 1), meaning:(11)$$\tau_{n,m}\left(x_{s},y_{s}\right)=\frac{\sqrt{{\left(x_{s}-x_{n}\right)}^{2}+{\left(y_{s}-y_{n}\right)}^{2}}-\sqrt{{\left(x_{s}-x_{m}\right)}^{2}+{\left(y_{s}-y_{m}\right)}^{2}}}{c}$$

Furthermore, the factor $R_{n,m}\left(\tau_{n,m}\left(x_{s},y_{s}\right)\right)$ is the cross-correlation between the *n*th and the *m*th microphone of the network and it is calculated by:(12)$$R_{n,m}\left(\tau_{n,m}\left(x_{s},y_{s}\right)\right)=\frac{1}{2\pi }\int_{-\infty }^{\infty }\Psi_{n,m}^{PHAT}\left(\omega \right)X_{n}\left(\omega \right)X_{m}^{*}\left(\omega \right)e^{i\omega t}d\omega$$
where $\Psi_{n,m}^{PHAT}$ is the weighting function in the frequency domain and the phase transform is:(13)$$\Psi_{n,m}^{PHAT}\left(\omega \right)=\frac{1}{|X_{n}\left(\omega \right)X_{m}^{*}\left(\omega \right)|}$$

Written in its discrete form, $R_{n,m}\left(\tau \right)$ is:(14)$$R_{n,m}\left(\tau_{n,m}\left(x_{s},y_{s}\right)\right)=\sum_{l=1}^{L}\frac{X_{n}\left(\omega \right)X_{m}^{*}\left(\omega \right)}{|X_{n}\left(\omega \right)X_{m}^{*}\left(\omega \right)|}e^{i\omega_{l}\tau_{n,m}\left(x_{s},y_{s}\right)}$$

It is important to note that the output of this technique is not a reconstructed signal, but the power of the signal at a given point (x,y). When carrying out this technique over different points, a 2-dimensional steered power response is provided, which be considered as the acoustic energy map for our purposes.

### 3.4. Phase-Based Binary Masking (PBM)

Phase-based Binary Masking (PBM) [24,25] is not an actual beamformer, but it has a similar application for the acoustic mapping. This technique employs the first stage of the DAS beamformer, such that the information arriving from a given steered position is aligned in all nodes. Then, the average phase difference ${\left|\varphi \right|}_{\omega }$ is calculated between all of the possible pairs of *m*th and *n*th nodes in the network, for all ω of the input signal.
(15)$${\left|\varphi \right|}_{\omega }=\frac{2}{M(M-1)}\sum_{m=1}^{M-1}\sum_{j=n+1}^{M}\left|\varphi_{m,\omega }-\varphi_{n,\omega }\right|$$
where $\varphi_{m,\omega }$ is the phase of the signal captured at node *m* at frequency ω after it being aligned by multiplying it by the weights in Equation (Equation 5).

Consequently, a phase difference threshold ($\varphi_{max}$) is then used to establish if ω is a frequency component of the reconstructed signal or not, effectively creating the following binary mask:(16)$$B\left(x,y,\omega \right)=\left\{\begin{array}{cc} 1, & {\;\mathrm{if}\;|\varphi |}_{\omega }\leq \varphi_{max}\\ 0, & \;\mathrm{otherwise} \end{array}\right.$$

The output signal is then calculated as:(17)$$S\left(x,y,\omega \right)=B\left(x,y,\omega \right)*X_{1}\left(\omega \right)$$

## 4. Simulation Results and Analysis

In this section, the impact of synchronization errors in the generation of the acoustic energy map are studied to determine the required precision levels. The hypothesis presented is that if there is an error in synchronization, the shape of the acoustic energy map will be modified which will result in an error of the position of the acoustic sources.

For this analysis, an acoustic source is simulated on a known fixed point (x,y), as well as a WASN with four nodes on a rectangular distribution with the source located in the same plane. For simplicity, the position of the source is set at (x,y)=(0,0), so the distance between each microphone and the source is the same.

Then, the four previously described beamformers are used to reconstruct a signal at each given proof point. The obtained acoustic map can be represented in a 3D plot where the XY-plane is the physical plane of the WASN and the *Z*-axis represents the acoustic energy associated to each point. Another way to present the acoustic map is as a heat-map, where each proof points is presented with a determined color associated to its energy.

The characteristics of the simulated acoustic environment are:Sampling rate of the capturing nodes: $f_{s}=48,000$ HzSpeed of sound in air: $c=343\frac{m}{s}$Amplitude signal of the source = cos(2∗π∗ω∗t)Frequency of the source: ω=300 HzTime between samples: $\tau_{s}=\frac{1}{f_{s}}=0.0208$ msCoordinates of the source: (x,y)=(0,0)Coordinates of the nodes:−$\left(x_{1},y_{1}\right)=\left(-0.32,0.32\right)$−$\left(x_{2},y_{2}\right)=\left(0.32,0.32\right)$−$\left(x_{3},y_{3}\right)=\left(-0.32,-0.32\right)$−$\left(x_{4},y_{4}\right)=\left(0.32,-0.32\right)$

As it was mentioned, the time-dependent energy function is proportional to the squared amplitude of the time-dependent signal function, for simplicity and without losing generality, the proportionality constant is proposed to be 1, so that the energy function is:(18)$$E\left(x,y,t\right)={∥S\left(x,y,t\right)∥}^{2}$$

The original amplitude and energy for the source are represented in Figure 9:

The signals captured by the four nodes of the simulated WASN are shown in Figure 10. In this image, all of the signals are overlapped since all of them have the same arrival time. In the synchronization analysis, one of this nodes will be artificially de-synchronized so that the signal is delayed a certain amount of samples.

The acoustic maps obtained by applying each of the beamformers to the simulated input signals are shown in Figure 11, and their respective heatmaps are shown in Figure 12.

The proof point with the largest energy is (x,y)=(0,0). The energy obtained with the reconstructed signal in this proof point with the DAS, MVDR and PBM beamformers are shown in Figure 13. Additionally, the 2D steered power response provided by SRP-PHAT is also shown.

To analyze the performance on the generation of the acoustic energy map, the total energy in a period of time will be calculated. In the simulated signal, the acoustic energy in a period of time can be calculated by the integral and approximated by the trapezoidal method with the following expression:(19)$$\int_{t=0}^{t=t_{f}}S\left(x,y,t\right)dt\approx \frac{t_{f}}{2N}\sum_{n=1}^{N}\left(S\left(x,y,t_{n}\right)+S\left(x,y,t_{n+1}\right)\right):=\sum_{n=1}^{N}S\left(x,y,t\right)$$
where S(x,y,t) is the simulated signal of the source, $t_{f}$ is the time length of the signal, and *N* is the size of the signal in number of samples. The calculated sum for the simulated signal is:(20)$$\frac{t_{f}}{2N}\sum_{n=1}^{N}\left(\cos^{2}\left(2*\pi *\omega *t_{n}\right)+\cos^{2}\left(2*\pi *\omega *t_{n+1}\right)\right)=0.0049791$$

While the calculated sum for the reconstructed energy in proof point (0,0) is:(21)$$\sum_{n=1}^{N}S_{DAS}\left(0,0,t\right)=\sum_{n=1}^{N}S_{MVDR}\left(0,0,t\right)=\sum_{n=1}^{N}S_{PBM}\left(0,0,t\right)=0.0049861$$

This shows that the energy shown in the acoustic map is quite close to the energy of the original simulated signal.

Furthermore, the results show that the beamformers generate an acoustic energy map with an important amount of energy centrally located around the position where the simulated signal originated. However, there are some considerations that need to be addressed:There are differences in the shape of the acoustic energy map obtained with each beamformer. In this regard, some are more robust against noise and reverberation (such as MVDR), while others are more efficient in terms of processing time and simplicity on their implementations (such as DAS and PBM). The main objective of this analysis is in terms of performance in the presence of errors in synchronization, thus, the analysis of these other elements are left for future work.The value of the calculated energy from the reconstructed signal in the proof point (x,y)=(0,0) for DAS, MVDR and PBM has a small 0.14% error in comparison to the energy of the original input signal. This can be considered as evidence of good performance of these beamformers when it comes to reconstructing the source of interest in the proof point that matches the simulated position of the source.DAS, MVDR and SRP-PHAT provide moderately high energy values in proof points that are not at the position of the source. The PBM beamformer finds energy only in the simulated position of the source.The PBM beamformer seems to be the most accurate for locating the position of the source. However, this technique requires a parameter to be set *a-prori*, thus, there is an extra stage of calibration for the implementation of this technique.The SRP-PHAT beamformer, like DAS and MVDR, finds energy in points where there is not acoustic energy from the source, but also encounters a pattern that is not representing an expected effect of the beamformer.

### Simulation of Synchronization Error

The acoustic energy map model for the four proposed beamformers results in the correct localization of the simulated acoustic source. Now, their performance in the presence of error in synchronization will be analyzed.

It is important to remember that the beamformers rely on the proper synchronization of node network, which is not always the case with WASNs. To simulate this, a delay (established as a determined number of samples) is artificially inserted in the first simulated node, located in $\left(x_{1},y_{1}\right)=\left(-0.32,0.32\right)$, in the upper left position of the array, referred here as “sync-error node”.

A synchronization error implies that before the signal arrives to the delayed node, there is no acoustic signal, as shown in Figure 14: the delayed input signal bears a value of zero for the simulated delay time. The beamformers may have problems with this discontinuity in the time domain, which may be solved by applying a window (such as Hann or Hamming) to reduce its effects, but this analysis is left for future work.

This simulated synchronization error is then fed to the beamformers, and its impact is observed in the following figures, and quantified in the change in the shape of the acoustic energy map, in the position of the source and in the value of its reconstructed energy.

The DAS beamformer energy maps are shown in Figure 15. As it can be seen, the position of the maximum value of energy (and its surrounding “hill” or lobe) gets reallocated to a different position as the samples delay increases, shifting away from the sync-error node. In Figure 15a,b, this behavior is consistent up until the 60-sample delay; from then on, another lobe appears close to the sync-error node.

The “moving” trends of the DAS beamformer are also present in the MVDR beamformer, as it is shown in Figure 16. However, an important difference is that the size of the lobes are far smaller, providing better precision.

The reason why these trends occur is many-fold: these beamformers rely on the time-difference-of-arrival between nodes, and a delay adds to this, resulting in the source “moving”; they also compensate these time differences by artificially shifting the input signals, which bares a cyclical a nature, causing the “return” to the sync-error node. In any case, it can be seen that for DAS and MVDR the shape of the acoustic energy map changes if the nodes of the network aren’t synchronized. This results in a change of the estimated position of the source, determined by locating the maximum value of energy in the map.

As for the PBM beamformer, as it can be seen in Figure 17, with most simulated delays, the maximum energy value is in the (x,y)=(0,0) position. However, there are some simulated delays in which this is not the case.

As for the SRP-PHAT, since it does not calculate the actual energy (only the steered power), the heatmaps shown in Figure 18 aren’t technically acoustic energy maps. However, they can still be used for locating acoustic sources. It can be seen that the shape of the steered power response map doesn’t change much, even between 20 and 80 delay samples (reason which no other heatmaps is shown in Figure 18). Meaning, this technique is mainly used to only determine the most probable proof point where a single source may be located.

With these results it is possible to find the position of the acoustic source by locating the maximum value of energy. Thus, the next part of the analysis is to determine, if there is any, a relationship between the errors in synchronization and the position shift of the acoustic source. After finding the position of the acoustic source, the difference between the known position of the source $r_{s}$ and the obtained position of the source $r_{m}$ was calculated:(22)$$\Delta \vec{r}=\left|\vec{r_{s}}-\vec{r_{m}}\right|$$

The results of the simulation in 10 samples intervals are shown in Figure 19. It can be seen that there is a tendency in the DAS and MVDR beamformers to increase the localization error as the error in synchronization increases. The PBM and the SRP-PHAT beamformer present the lowest errors in localization, but the PBM beamformer generates errors in localization for particular values of delay (not with the tendency observed in DAS and MVDR). The SRP-PHAT beamformer has the most consistent performance, but given the noisiness of the shape of its steered response, it should only be used for acoustic source localization, not for the generation of an acoustic energy map.

As it can also be seen in Figure 19, there is a considerable “jump” in localization error over the 60-sample simulated delay. Thus, this is the upper delay value we used to establish a relationship between localization error and synchronization error. In Figure 20, this relationship is shown using all possible delays in the range of 1 to 60 samples:

As it can be seen, there is a near-linear relationship between localization error and synchronization for DAS and MVDR. Applying basic statistic fitting, these are the models that best describe this relationship:$\Delta {\vec{r}}_{DAS}=124.8\Delta t-0.0002519$, with $R^{2}=0.9933$$\Delta {\vec{r}}_{MVDR}=123.6\Delta t-0.0019950$, with $R^{2}=0.9844$
where Δt is the synchronization error in node 1.

It is important to note that MVDR has some values for Δt where $\Delta \vec{r}$ is much larger than the rest, thus, were treated as outliers for the fitting process. The resulting location-synchronization models for DAS and MVDR are shown in Figure 21:

There is a “step” behavior in Figure 21, which can be explained by the grid-type search that the acoustic energy map is based on. Additional simulations were carried out with a larger number of proof points, and it was observed that these “steps” were closer to each other, i.e., the grid is refined. The maximum values for synchronization errors didn’t change, but the processing time did increased considerably. Thus, there is a compromise to be made: a refined “step” behavior for a larger processing time.

As for the PBM and the SRP-PHAT beamformers, they tend to locate the acoustic source exactly in the position where it was simulated in a near-consistent manner across the simulated time-delays, having an error near to zero for the same limit of 60 samples of delay. Thus, no relationship model was calculated for these beamformers.

## 5. Experimental Results and Analysis

In this section, the recording methodologies proposed for the capture of acoustic signal and its application on the generation of the acoustic energy map will be analyzed. A WASN composed by the devices described in Section 2 was used to capture an acoustic source (a speaker emitting an acoustic sinusoidal wave of 300 Hz) placed in the center of the WASN. The network was built into a rectangular array with the dimensions presented in Section 4. The WASN and the capture node are shown in Figure 22a,b. The schematic representation of the WASN and the source is shown in Figure 23.

Each of the nodes captured a signal with the MEMS microphones. Then, a post-processing adjustment was applied for the generation of the acoustic energy model, as follows:**Retrieve data and timestamps adjustment**. Each node generates audio and time data files for the local recording methodologies: an audio file (in WAV format) for the captured signal and a text file with the timestamps generated during the recording. Afterwards, these files are retrieved by a processing unit. The four captured signals are shown in Figure 24a, without any post-processing, with their own timestamps and amplitude values. As it will be seen later, when the local methodologies for signal capture are implemented, each node begins recording at a different time. Thus, the timestamps of each of the recordings of the nodes are adjusted to begin at the time where all the timestamps are closest. Each recording has associated the time of the local clock, so it is also necessary to establish a global frame of reference for the time. All signals are normalized and adjusted to begin at t=0 as Figure 24b shows.**Sample selection**. With the adjusted timestamps, a segment of the full signal was selected for the generation of the acoustic energy map. A Hann window was applied to this segment to avoid frequency bleed-over effects when the signals are processed by all the proposed beamformers shown in Figure 25a,b.**Acoustic energy map generation**. The post-processed captured signals were processed by the beamformers to generate the acoustic energy map with the techniques described in Section 3. In this stage of the processing it is also possible to determine the position of the acoustic source by finding the maximum value on the acoustic energy map.

### 5.1. Synchronization Analysis

Given the location of the acoustic source, its signal should arrive to the each of the nodes of the WASN at approximately the same time. Thus, the analysis of synchronization can be based on the difference of the arrival times of the acoustic signal of the source to each node of the WASN. Additionally, it can also be based on the shape of the acoustic energy map generated by the signals captured by the WASN, compared to the ones obtained in the simulation without simulated synchronization errors, as well as to each other.

For the difference in arrival times analysis for each synchronization methodologies, the time of arrival of the signal of the source to each node was calculated and, by taking node 1 as reference, the differences of arrival times were calculated. In a perfectly synchronized WASN, with a perfectly placed acoustic source, the difference in arrival times between the reference node and all the other in the WASN should be zero. It is important to mention that synchronization errors may happen because of wireless interferences or mishaps in the TCP/IP hardware/software stack; the geometry of the array has no effect on the synchronization errors. Thus, each methodology will perform differently solely based on its characteristics and implementation. The results are shown in Figure 26.

These results show that the PTP-Local methodology has the lowest errors in synchronization, while the NTP-Local methodology has the largest errors in synchronization.

As for the map-shape analysis, in the following sub-sections (one for each synchronization methodology), two maps for each beamformer will be presented. The map on the left is representative of the best maps the beamformer provided given the synchronization methodology, and the map in the right representing the worst maps. The best and worst energy maps were obtained by capturing the signal of an acoustic energy source with a WASN as shown in Figure 22 and Figure 23. The evaluation process involved running each methodology to capture 30 samples of 60 s each. With these signals, in post-processing, the beamformers were applied to generate an acoustic energy map from each sample. The map that was the most similar to the real acoustic environment is presented here as the best map. The worst map is the one that presented the most discrepancy with the real acoustic environment.

#### 5.1.1. NTP-Local Methodology

The NTP-Local methodology generated the worst acoustic energy maps, which is expected since it obtained the largest amount of synchronization errors. Thus, in the worst cases, the beamformers did not generate an acoustic energy map that represented the nature of the acoustic environment. However, in the best cases where synchronization errors were low, the generated acoustic energy maps were able to locate a maximum of energy in the shape of an acoustic source.

For DAS, Figure 27a,b, the shape of the acoustic energy map doesn’t present a maximum value as a point source, instead it is distributed in a region (or lobe). For MVDR, Figure 28a,b, the energy map presents two regions where there seems to exists two acoustic sources, neither of which correspond to the real position of the acoustic source. For PBM, Figure 29a,b, the best case presented an almost point acoustic source, but not in the position of the acoustic source; in the worst case, several proof points presented a considerable amount of acoustic energy, none of which represented the acoustic source’s position. Finally for SPR-PHAT, Figure 30a,b, in the best map, the most likely point for the localization of the source is close to the real position; in the worst map, there are many possible points where the source can be located.

#### 5.1.2. PTP-Local Methodology

The PTP-Local methodology shows an improvement over the NTP-Local methodology in terms of the shape and position of the lobes in the acoustic energy map. It was the most consistent of the methodologies, for all of the beamformers, locating the acoustic source near its real position.

For DAS, Figure 31a,b, even in the worst map, the acoustic energy map always presented a lobe with a localized maximum acoustic energy. For MVDR, Figure 32a,b, it presents smaller lobes (which are desirable), centered around a location that is close to the real location of the source. For PBM, Figure 33a,b, it presents the same problem as it did with the NTP-Local methodology: when there is a synchronization error, the beamformer finds energy in many proof points of the acoustic energy map. For SPR-PHAT, Figure 34a,b, it also performs similarly as with the NTP-Local methodology.

#### 5.1.3. PTP-Remote Methodology

The PTP-Remote methodology shows that when the WASN is correctly synchronized, the acoustic energy maps obtained are close in precision to the ones obtained with the PTP-Local methodology. However, when the WASN presents synchronization errors, the maps obtained have a performance with the same problems as the ones seen in the NTP-Local methodology.

It is important to mention that, although the worst maps are similar to the worst maps obtained using NTP-Local methodology for DAS, Figure 35a,b, in the case of MVDR, Figure 36a,b and PBM, Figure 37a,b, their best maps are similarly localized compared to their respective best maps obtained using the PTP-Local methodology, except for SPR-PHAT, Figure 38a,b. Additionally, acquiring the audio data is more practical using the PTP-Remote methodology compared to the other two methodologies (which require manual extraction from each node).

## 6. Discussion

In terms of precision in the localization of a simulated acoustic source, PBM tends to find high energy values only in the simulated position of the source, which is the ideal expected behavior. However, it is important to note that PBM requires a parameter (the phase difference threshold) to be established prior to the generation of the acoustic energy map; the rest of the beamformers do not require any *a-priori* calibration. As for SRP-PHAT, although it does not technically create an acoustic energy map, the location with the highest steered power matches the location of the simulated acoustic source even when synchronization error are inserted. And as for DAS and MVDR, although they present larger localization errors when synchronization errors are simulated, the relationship between these errors is near-linear. This insight can be very valuable since this relationship can be used to establish a type of precision threshold given an expected level of synchronization errors which, in turn, can be estimated from the nature of a given application scenario. Thus, from the findings in the experimental analysis, a user can know beforehand what type of localization errors are bound to happen using a WASN to create an acoustic energy map.

Interestingly, DAS and MVDR provide very similarly shaped acoustic energy maps, and share similar tendencies of shifts of the location of the lobe above the simulated acoustic source, displacing it away from the sync-error node up to a simulated synchronization error of 60 samples. This can be explained by the geometrical characteristics of the array and its relationship to the wavelength of the signal. The analysis of the nature of the beamformers and its implications to the acoustic energy map generation are left for further studies.

The experimental analysis show that an acoustic energy map can be generated using any of the described synchronization methodologies. The NTP-Local and the PTP-Remote methodologies provide the largest synchronization errors, while the PTP-Local methodology presented the lowest synchronization errors, as well as the lowest variance. Additionally, when an acoustic source was located in the center of the array, where the difference in arrival times between nodes should close to 0, the PTP-Local methodology provided the lowest values and variance. In the simulation analysis, 60 samples was found to be a type of upper threshold of synchronization error; with higher values, a big “jump” in localization errors was observed with DAS and MVDR. A synchronization error of 60 samples, sampled at 48 kHz, is equivalent to 1.25 ms. Connecting this with the observed mean difference of arrival times to the reference capture node ($mean_{1-2}=0.0958$ ms, $mean_{1-3}=-0.1555$ ms, and $mean_{1-4}=0.3035$ ms) when using the PTP-Local methodology, it can be deduced that it provides synchronization errors well below the aforementioned threshold and, thus, well suited to calculate acoustic energy maps with any of the described beamformers.

Having calculated an acoustic energy map using all the described beamformers along with the described synchronization methodologies, the position of their maximum value was found and considered as the estimated position of the acoustic source. The localization errors in the experimental scenario are shown in Table 3. The statistical measures presented are the mean and the Median Absolute Deviation (MAD) [26], which is a distribution-free measure of dispersion. These quantities are used as a comparison factor between methodologies.

As it can be seen, these localization errors are considerably larger than the ones found in the simulation analysis. This is expected since a real acoustic scenario has more variables that were not simulated (reverberation, other interferences, etc.). Additionally, the simulated synchronization errors were inserted into just one node (the sync-error node), while in the experiment scenario all nodes are affected by such issues. Finally, the speaker used in the experiments is not a point acoustic source, as was the case in the simulation. However, given all these factors, it is important to note that since these acoustic maps were generated from signals captured using a WASN, the placement of the nodes are not limited to any tethering, thus, are easily positioned anywhere in the acoustic scenario, such as in a living room. Considering this, the observed localization errors are reasonably low, with PBM using the PTP-Local methodology providing the lowest mean error. Interestingly, SRP-PHAT provides the lowest variance (also using the PTP-Local methodology), however, it provides the largest mean error, thus, it may not be as suitable as PBM.

## 7. Conclusions

In this paper, three different synchronization methodologies were used to generated an acoustic energy map applying four popular beamforming techniques: Delay-and-Sum (DAS), Minimum Variance Distortionless Response (MVDR), Steered-Response Power with Phase Transform (SRP-PHAT), and Phase-based Binary Masking (PBM).

The synchronization methodologies were conformed by: (1) a WASN capture protocol (either local or remote) used to capture a multi-channel audio recording from the environment; and (2) a synchronization protocol, which could be either Network Time Protocol (NTP) or Precision Time Protocol (PTP). The beamforming techniques were then applied to the captured signals, to measure the energy coming from a given series of proof points (arranged in a grid) to generate the acoustic energy map.

A simulation was carried to, first, validate this proof of concept, and to characterize the behavior each beamforming technique against a simulated synchronization error.

Furthermore, an experimental scenario was used to test the synchronization methodologies. The devices used for capture nodes in the WASN were conformed by a Raspberry 4B+ as a processing unit and a MEMS microphone breakout-board. This configuration was chosen because of its low cost, accessibility and straightforward implementation.

With the results obtained by simulation and experiments, the following conclusions can be stated:In the simulation, SRP-PHAT and PBM showed higher robustness against synchronization errors. DAS and MVDR were more sensitive, but a near-linear relationship between localization errors and synchronization errors was found which can be used for ease of expectation of the user.In the experimental scenario, from a subjective point of view, MVDR and PBM provided an acoustic energy map close to what was expected, with either a lobe or point over the acoustic source location; SRP-PHAT and DAS provided unexpected maps. While PBM requires a parameter to be calibrated beforehand, it generates a more precise acoustic energy map. And, while MVDR generates a less precise acoustic energy map, it is robust against environmental elements (such as noise and interferences) and does not require any a-priori parameter calibration.The PTP-Local and PTP-Remote methodologies are the most suitable for acoustic energy mapping using a WASN. The PTP-Local methodology has the best performance in terms of synchronization errors, but the process to capture, retrieve and process the data collected is more tedious than the PTP-Remote methodology, since it requires the user to manually extract the recording from the capture node. The PTP-Remote methodology is less reliable in terms of synchronization (with a higher mean error value). However, the captured signals can still be used to provide a close-to-precise acoustic energy map (with either PBM or MVDR), while being easier to implement (since no synchronization agent is required to run locally in the capture nodes), and the recordings are directly streamed to the server (no manual extraction necessary).

For future work, other synchronization protocols are to be explored. Additionally, other characterizations are to be carried out, such as: robustness against noise and reverberation, processing time, implementation complexity, etc. Finally, the effect of different windows (other than Hann, used in this work) to reduce time-domain discontinuities is to be characterized.

## Figures and Tables

**Figure 1 sensors-23-04645-f001:**
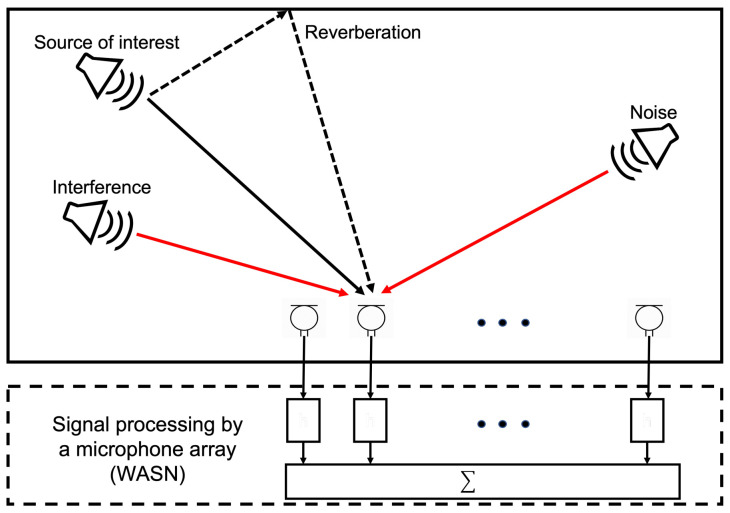
Acoustic signal analysis.

**Figure 2 sensors-23-04645-f002:**
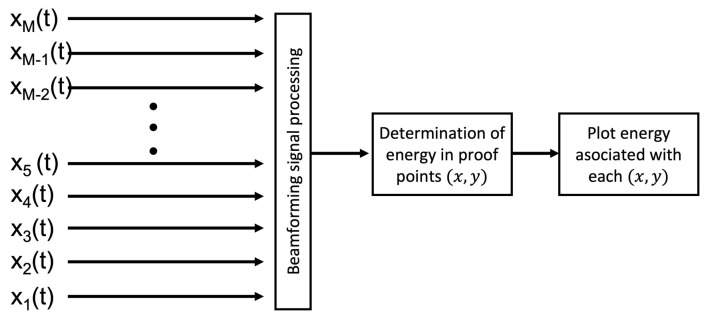
Methodology for acoustic energy mapping.

**Figure 3 sensors-23-04645-f003:**
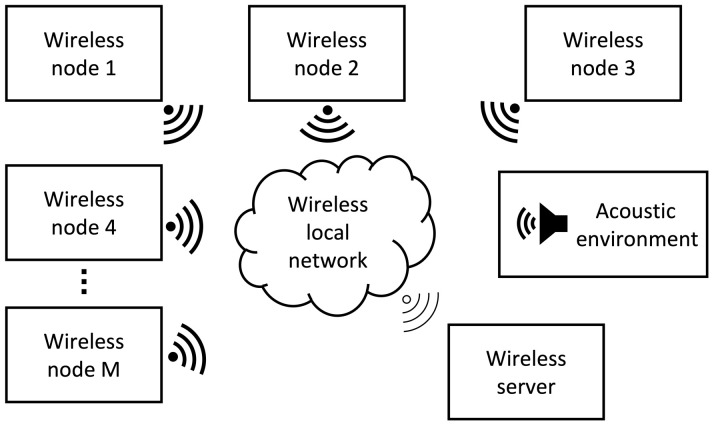
Wireless Acoustic Sensor Network.

**Figure 4 sensors-23-04645-f004:**
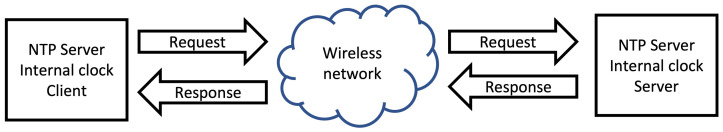
NTP.

**Figure 5 sensors-23-04645-f005:**
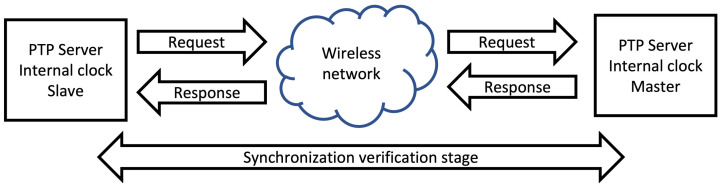
PTP.

**Figure 6 sensors-23-04645-f006:**
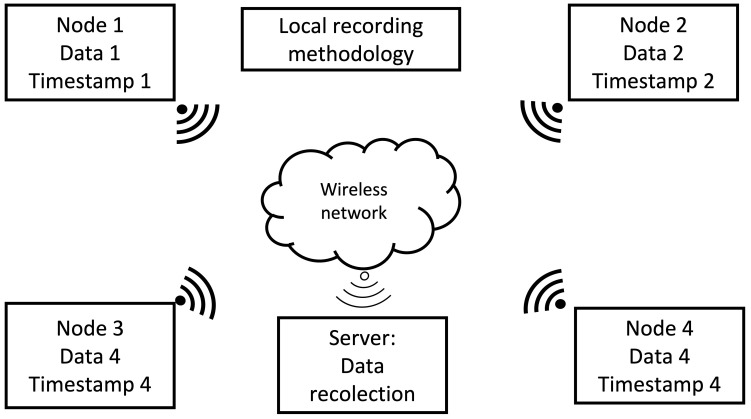
Local recording methodology.

**Figure 7 sensors-23-04645-f007:**
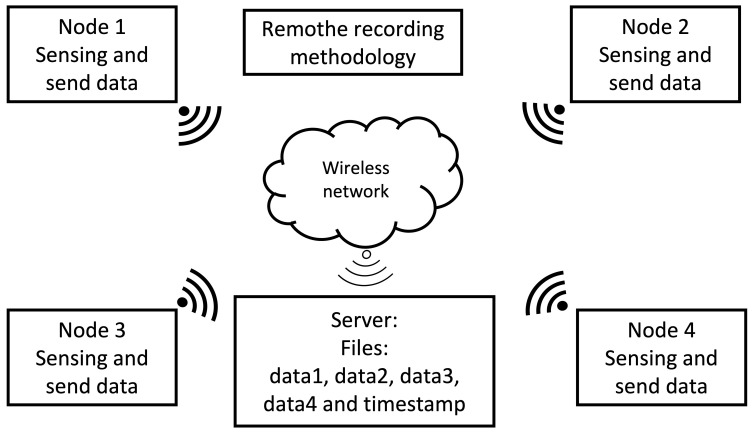
Remote recording methodology.

**Figure 8 sensors-23-04645-f008:**
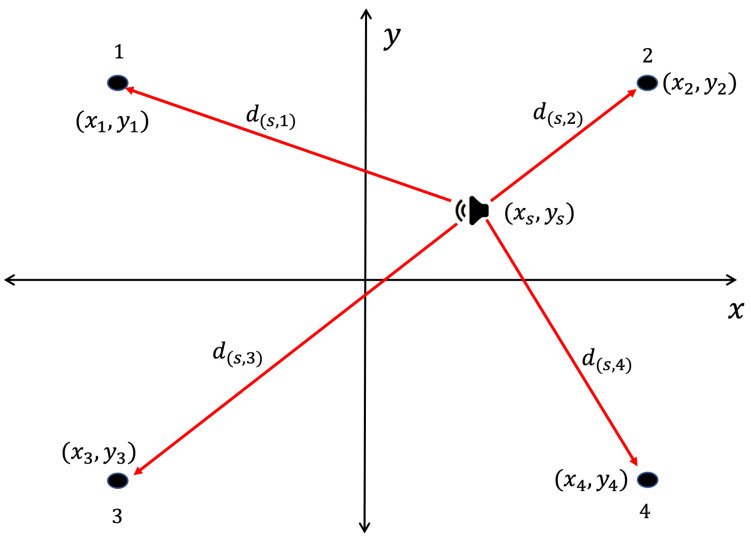
Distance between acoustic source and the nodes of a 4 nodes WASN process.

**Figure 9 sensors-23-04645-f009:**
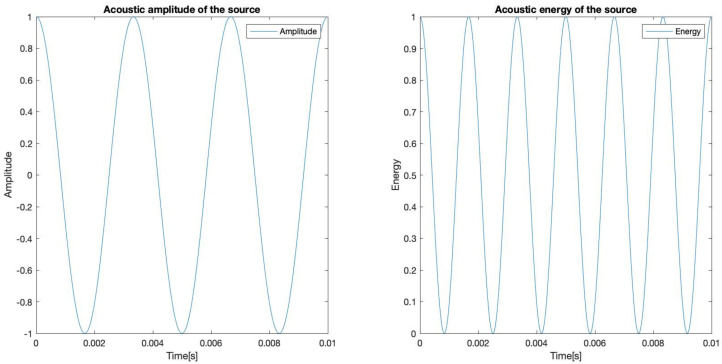
Original amplitude and energy signal of the source.

**Figure 10 sensors-23-04645-f010:**
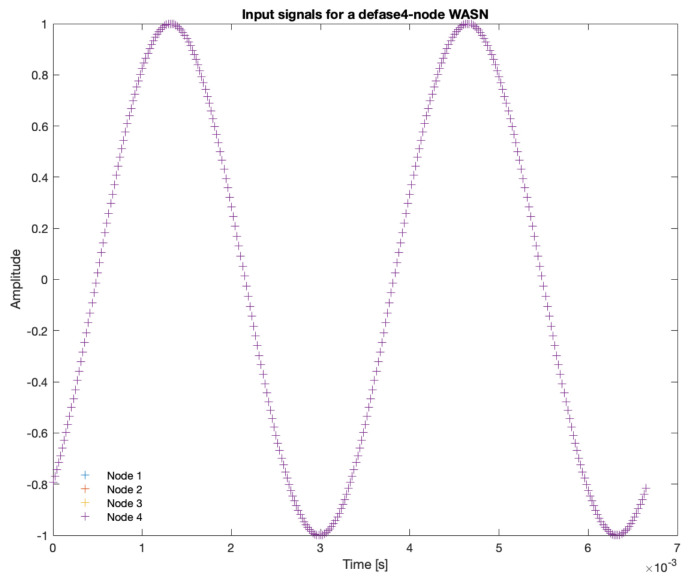
Input signals of the four nodes of the WASN.

**Figure 11 sensors-23-04645-f011:**
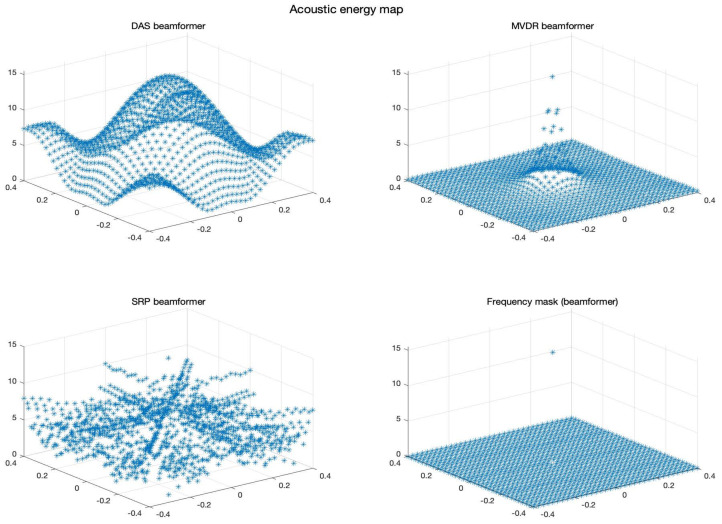
Acoustic energy map with: DAS, MVDR, SRP-PHAT and PBM beamformers.

**Figure 12 sensors-23-04645-f012:**
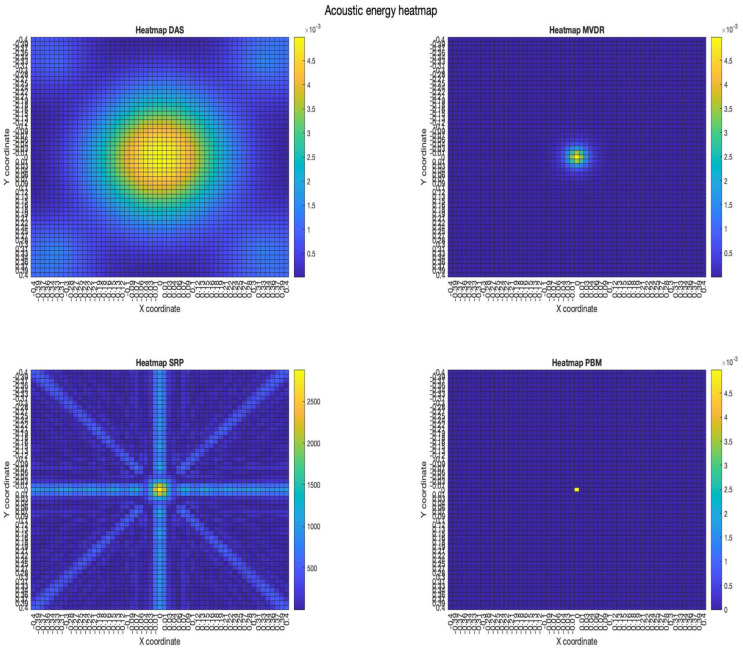
Heatmap of acoustic energy for beamformers: DAS, MVDR, SRP-PHAT and PBM.

**Figure 13 sensors-23-04645-f013:**
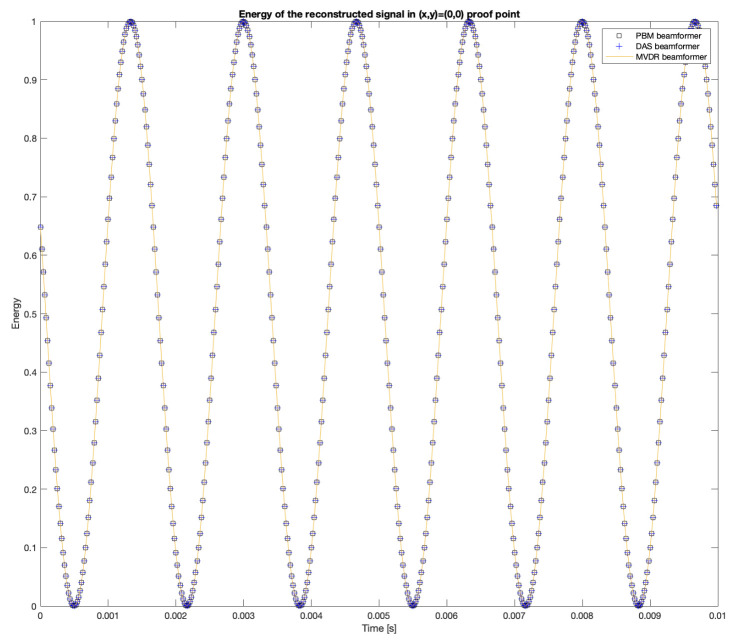
Energy of the reconstructed input signal at (x,y)=(0,0).

**Figure 14 sensors-23-04645-f014:**
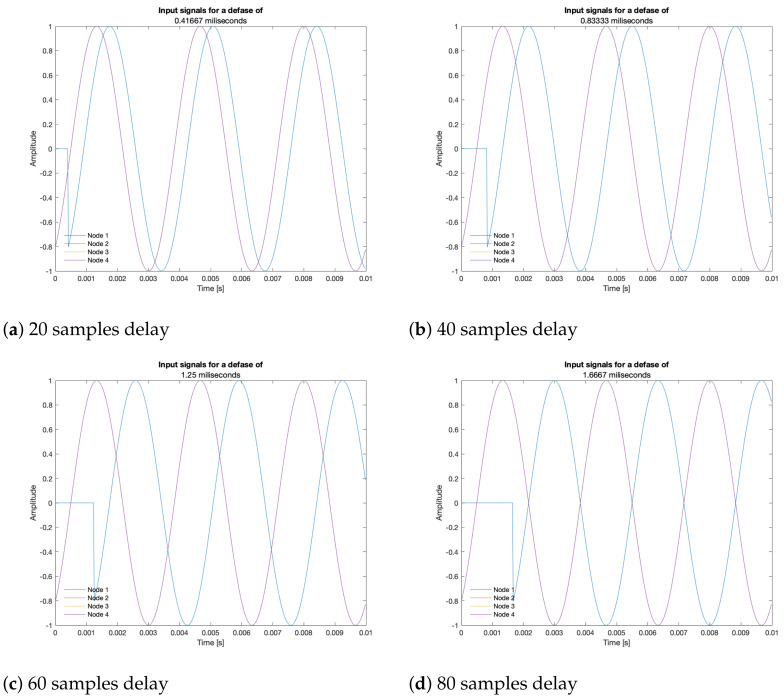
Input signals delayed by 20 samples intervals.

**Figure 15 sensors-23-04645-f015:**
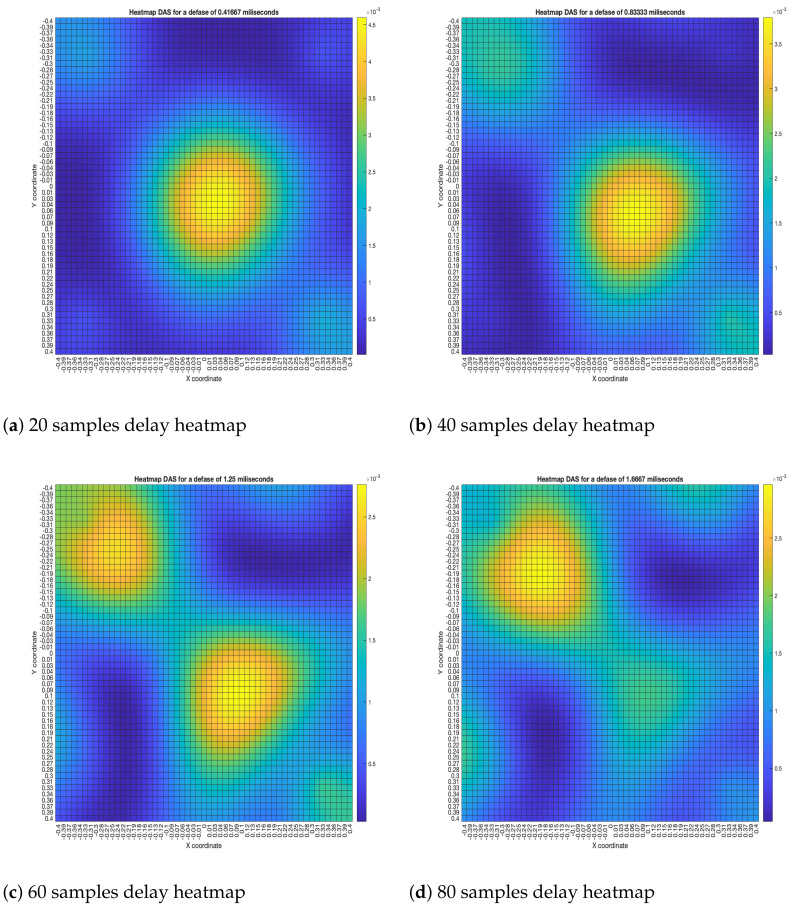
DAS energy heatmap for the simulated delay in capture.

**Figure 16 sensors-23-04645-f016:**
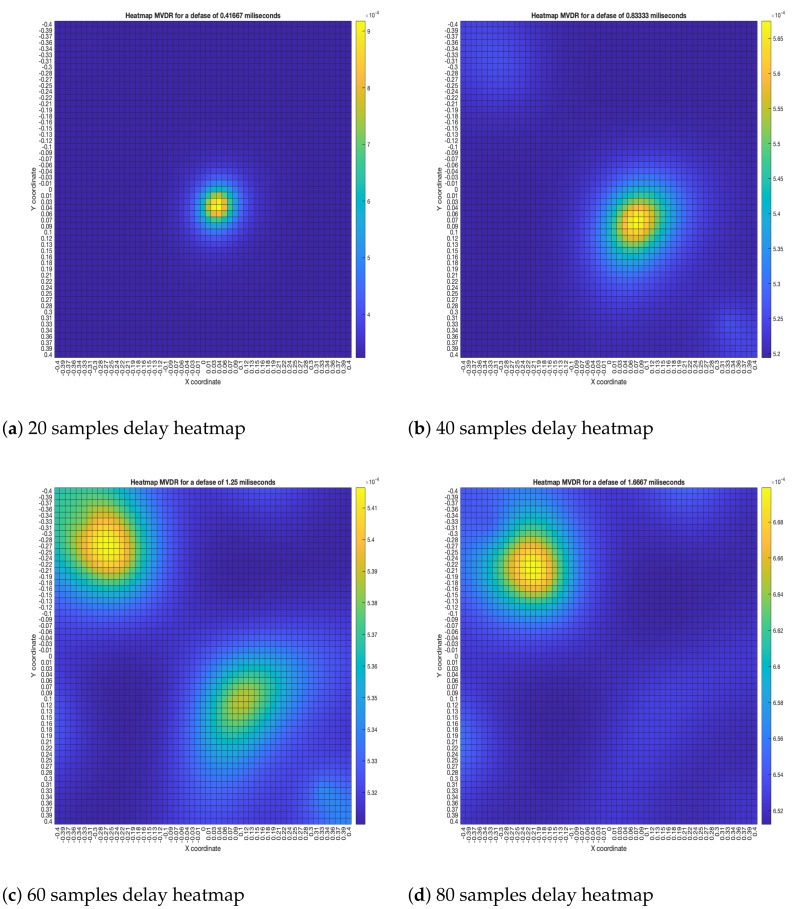
MVDR energy heatmap for the simulated delay in capture.

**Figure 17 sensors-23-04645-f017:**
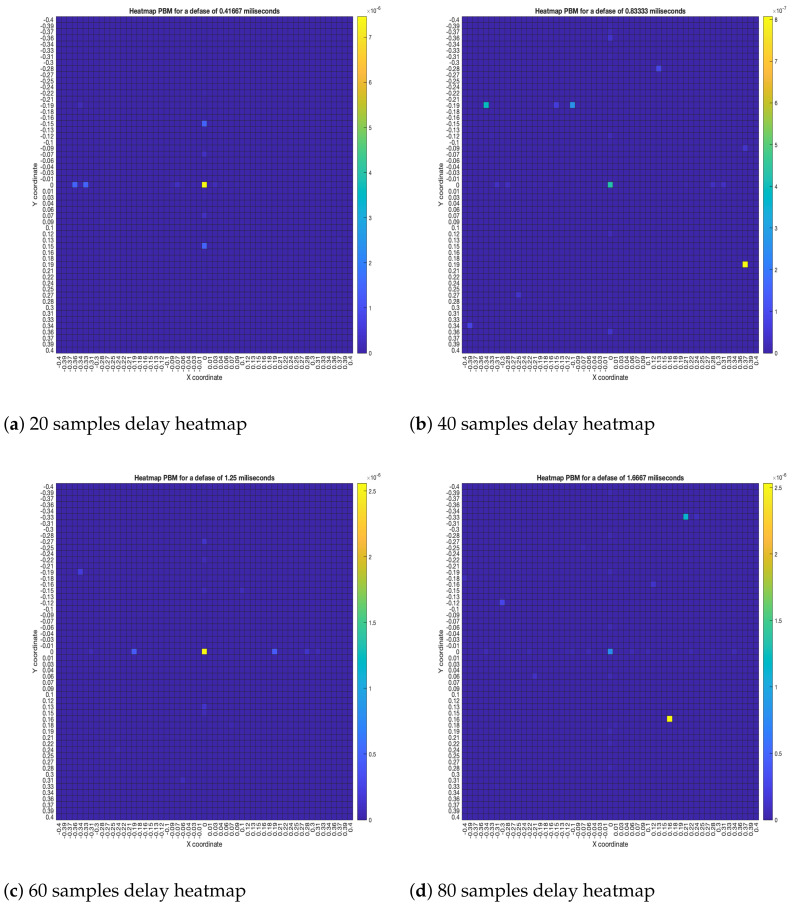
PBM energy heatmap for the simulated delay in capture.

**Figure 18 sensors-23-04645-f018:**
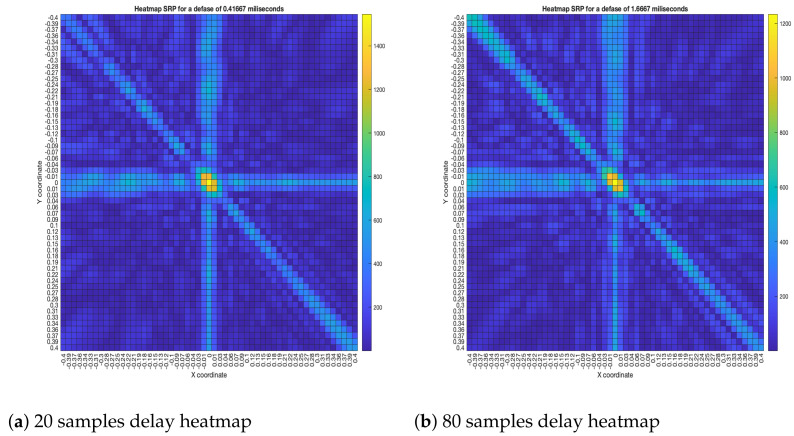
SPR-PHAT energy heatmap for the simulated delay in capture.

**Figure 19 sensors-23-04645-f019:**
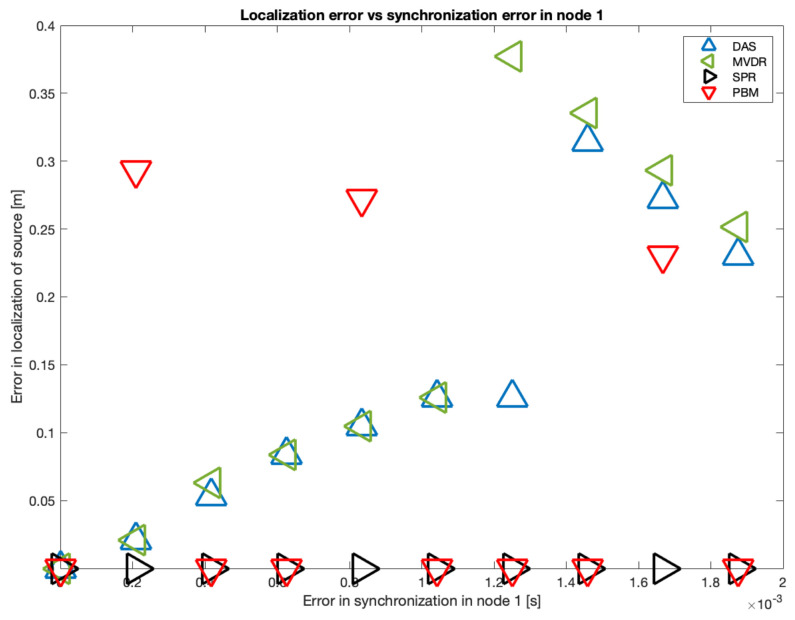
Localization errors for a source in (x,y)=(0,0) and errors in synchronization in node 1.

**Figure 20 sensors-23-04645-f020:**
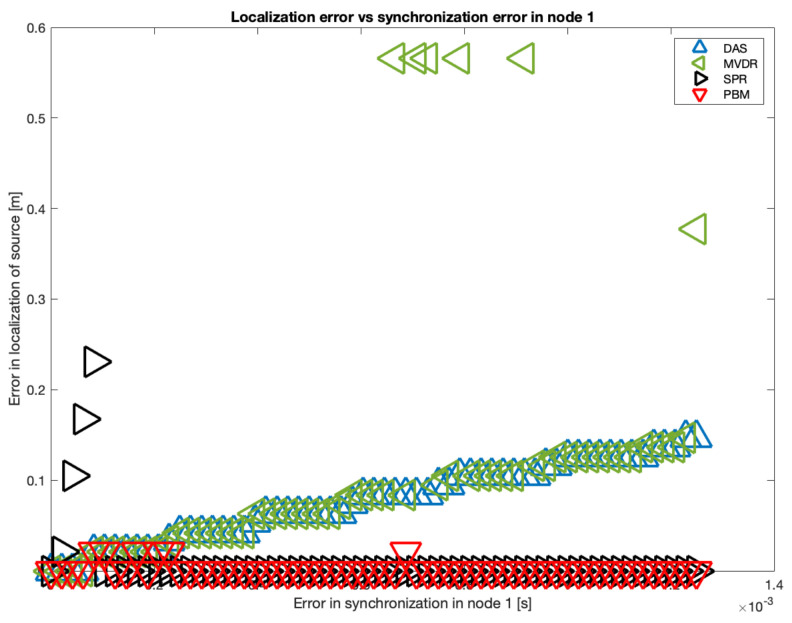
Errors in localization for all beamformers.

**Figure 21 sensors-23-04645-f021:**
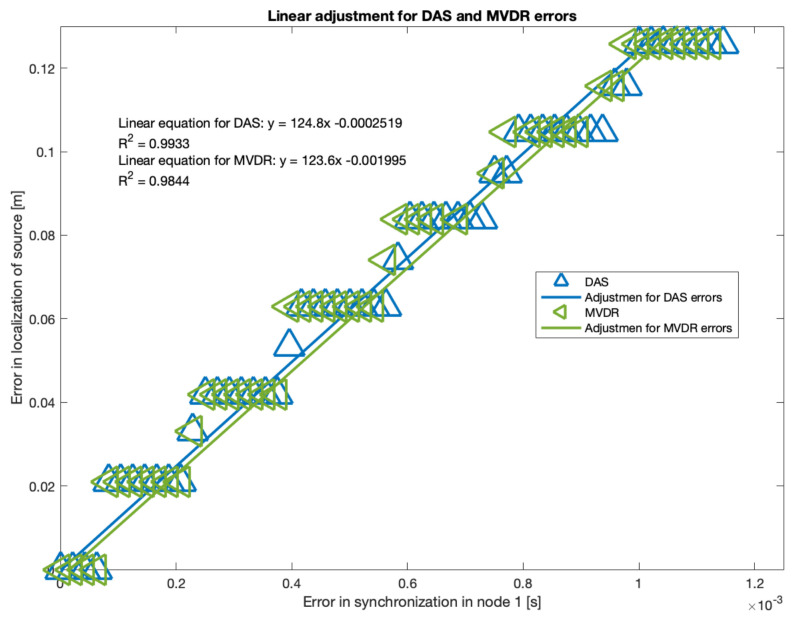
Fitting for the DAS and MVDR beamformer errors in localization.

**Figure 22 sensors-23-04645-f022:**
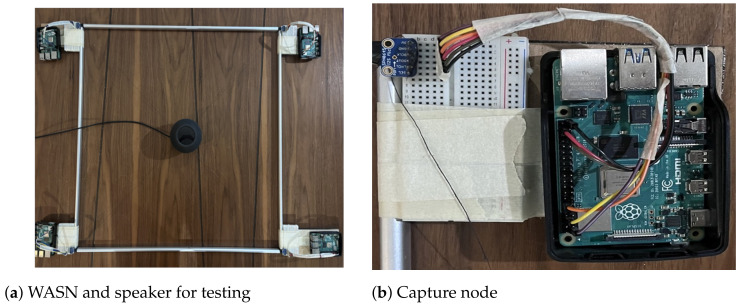
Experimental design for testing.

**Figure 23 sensors-23-04645-f023:**
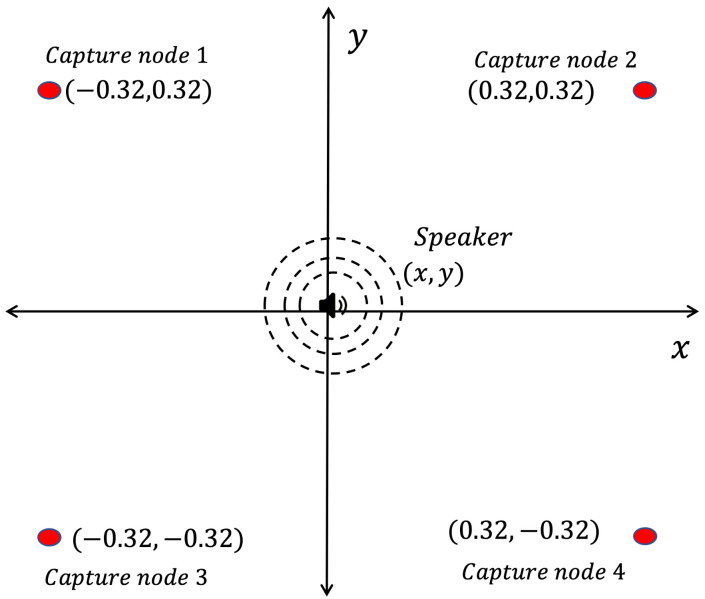
Experimental array for the evaluation of the methodologies.

**Figure 24 sensors-23-04645-f024:**
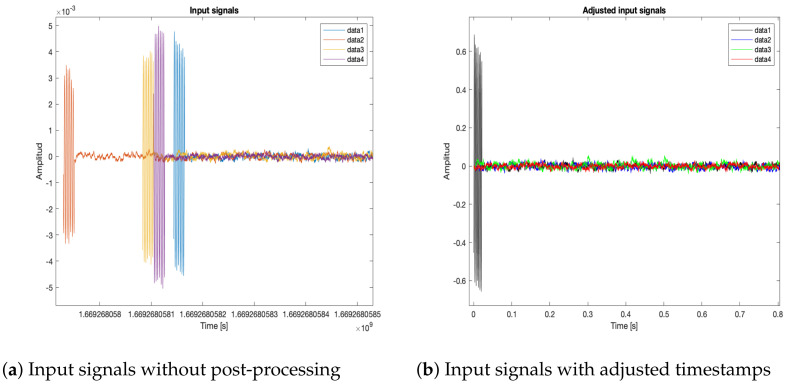
Retrieving and adjusting of audio data signals.

**Figure 25 sensors-23-04645-f025:**
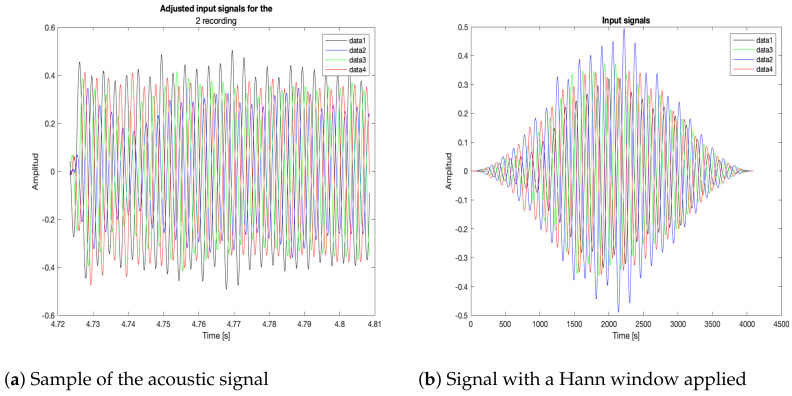
Post-processing of the acoustic input signals.

**Figure 26 sensors-23-04645-f026:**
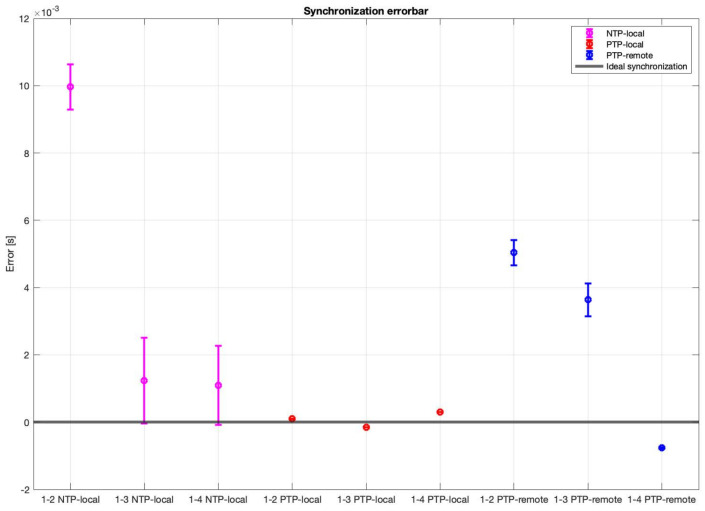
Synchronization errors, as boxplots, for each node and each methodology.

**Figure 27 sensors-23-04645-f027:**
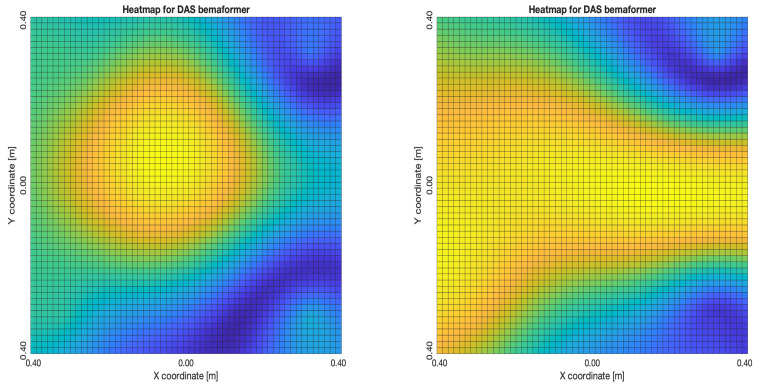
Acoustic energy maps obtained with DAS (left: best, right: worst).

**Figure 28 sensors-23-04645-f028:**
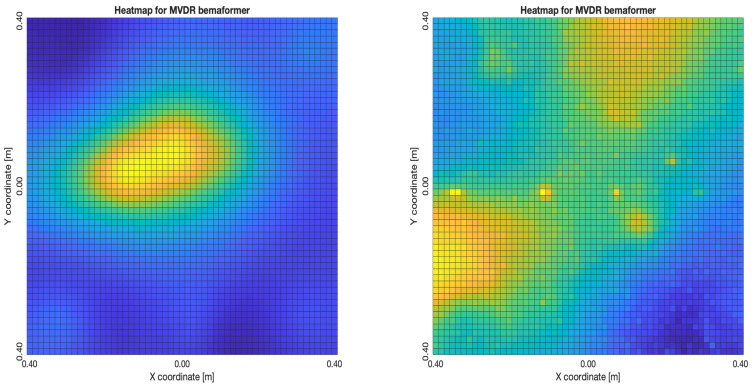
Acoustic energy maps obtained with MVDR (left: best, right: worst).

**Figure 29 sensors-23-04645-f029:**
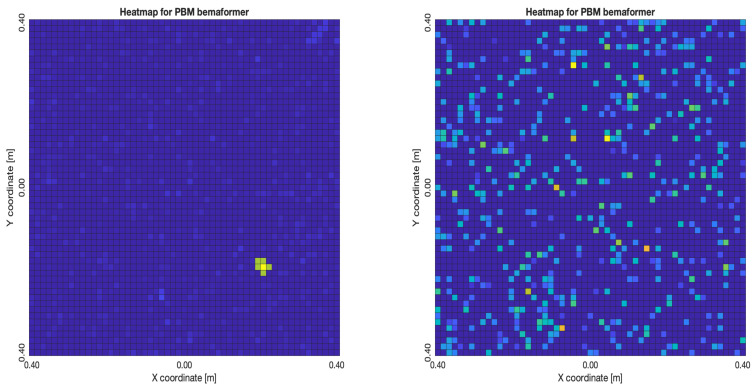
Acoustic energy maps obtained with PBM (left: best, right: worst).

**Figure 30 sensors-23-04645-f030:**
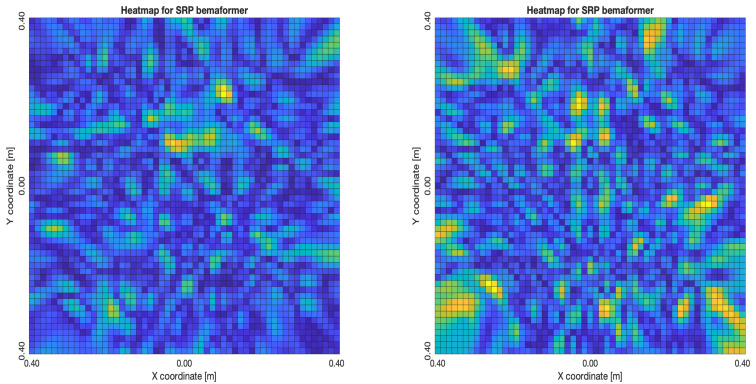
Acoustic energy maps obtained with SRP-PHAT (left: best, right: worst).

**Figure 31 sensors-23-04645-f031:**
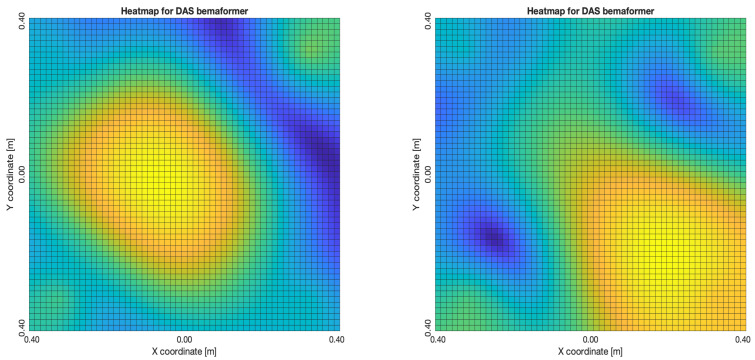
Acoustic energy maps obtained with DAS (left: best, right: worst).

**Figure 32 sensors-23-04645-f032:**
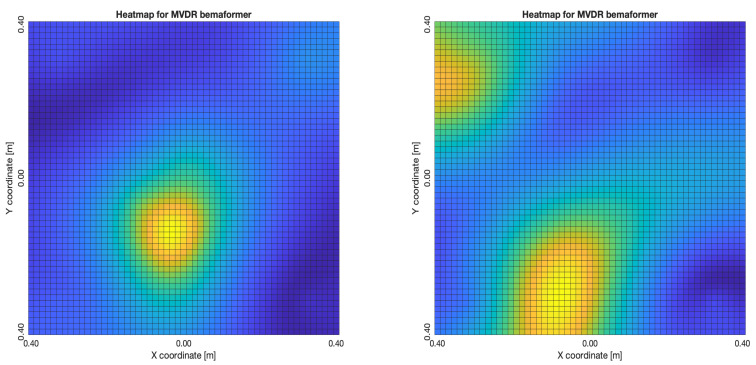
Acoustic energy maps obtained with MVDR (left: best, right: worst).

**Figure 33 sensors-23-04645-f033:**
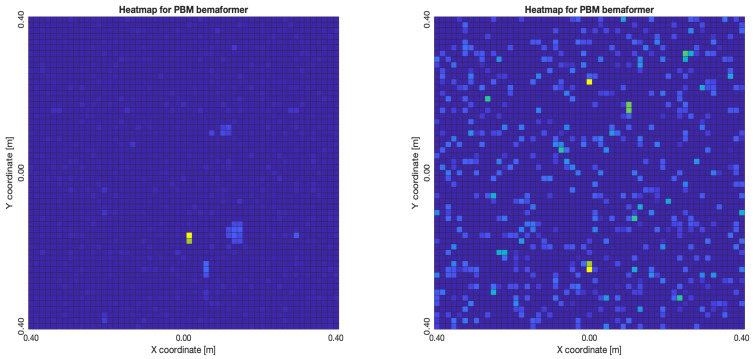
Acoustic energy maps obtained with PBM (left: best, right: worst).

**Figure 34 sensors-23-04645-f034:**
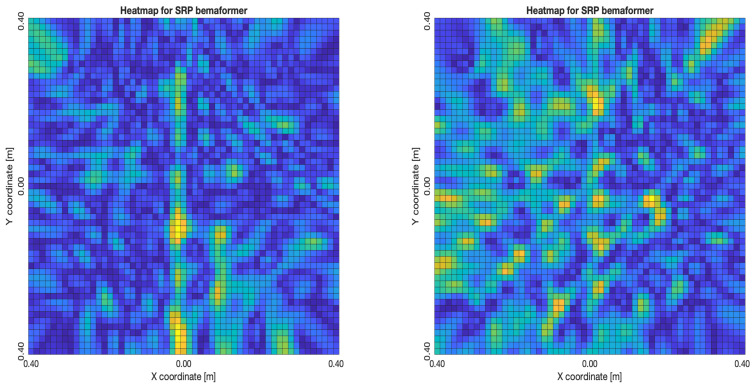
Acoustic energy maps obtained with SRP-PHAT (left: best, right: worst).

**Figure 35 sensors-23-04645-f035:**
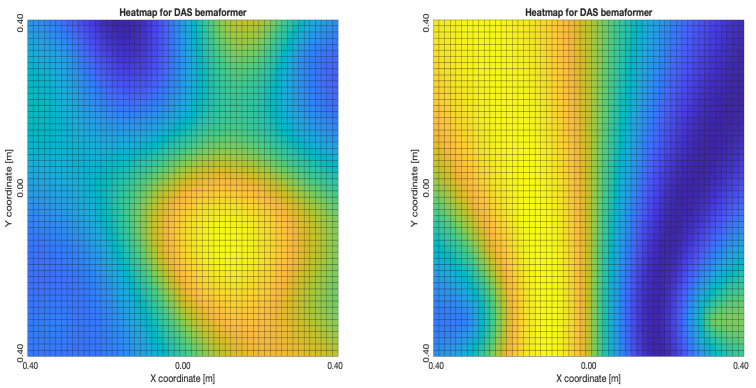
Acoustic energy maps obtained with DAS (left: best, right: worst).

**Figure 36 sensors-23-04645-f036:**
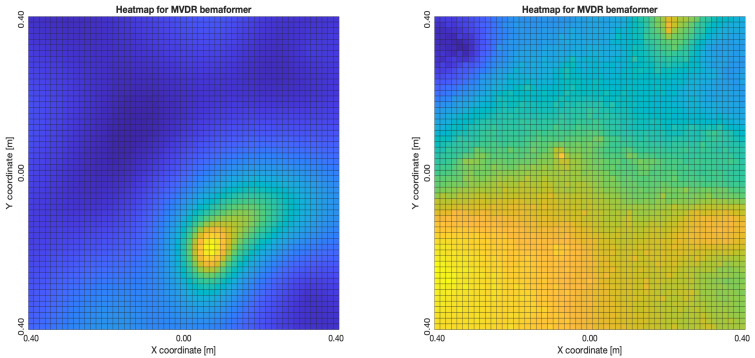
Acoustic energy maps obtained with MVDR (left: best, right: worst).

**Figure 37 sensors-23-04645-f037:**
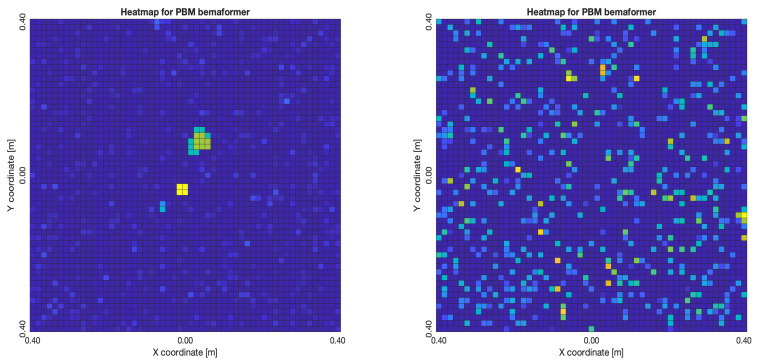
Acoustic energy maps obtained with PBM (left: best, right: worst).

**Figure 38 sensors-23-04645-f038:**
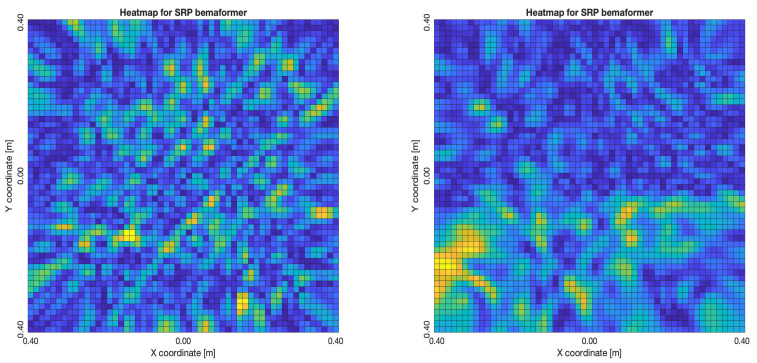
Acoustic energy maps obtained with SRP-PHAT (left: best, right: worst).

**Table 1 sensors-23-04645-t001:** Processing device characteristics.

Model	Processor	RAM	Conectivity
Raspberry Pi 4B+	Broadcom BCM2711, Quad core 64 bit @ 1.5 GHz	1 GB	2.4 GHz and 5.0 GHz IEEE 802.11ac wireless, Bluetooth 5.0

**Table 2 sensors-23-04645-t002:** Sensor characteristics.

Model of Microphone	Sensitivity	Signal to Noise Ratio	Output Interface
Knowles, I2S Output Digital Microphone: SPH0645LM4H-B	−26 dBFS	65 dB(A)	I2S

**Table 3 sensors-23-04645-t003:** Localization error analysis for each methodology and each beamformer.

Methodology	Localization Error [m]	DAS	MVDR	SRP-PHAT	PBM
NTP-Local	Mean	0.3429	0.3224	0.2814	0.3198
	MAD	0.1013	0.1182	0.0823	0.1359
PTP-Local	Mean	0.2585	0.2533	0.3046	0.2151
	MAD	0.1069	0.1126	0.0949	0.1247
PTP-Remote	Mean	0.2461	0.2922	0.3067	0.2803
	MAD	0.0930	0.1005	0.1015	0.1021

## Data Availability

Acoustic mapping code for simulated signals and real signals captured by a WASN of 4 nodes: https://github.com/emilianounzueta/acoustic-energy-mapping.

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
