# Peer review of "On the Challenges of Acoustic Energy Mapping Using a WASN: Synchronization and Audio Capture"

_sensors, 2023, doi:10.3390/s23104645_

Round 1
Reviewer 1 Report
drawings are very small. The study is interesting but needs to be improved.
"The analysis of the acoustic scene can be used in multiple applications: urban moni- 18 toring [1], bio-localization [2,3], rescue work in catastrophe situations [4], domestic smart 19 systems, etc. In such scenarios, the acoustic information of the environment (sources of 20 interest, noise, interference and reverberation) is obtained by processing acoustic signals 21 captured by a monitoring system [5,6] as shown in figure 1" this comparison and critical analysis is very general. Make a more detailed and complete analysis.
The bibliography should be expanded. Authors should use different high-ranking journals. It is necessary to show who else is engaged in these studies. Why the problem was not solved. Authors need to pay more attention to mathematical models and algorithms. Mathematical formulas are given without a normal explanation. Show also at the beginning what tasks they set and how they solved them. This is not in the article. The examples poorly reflect the problem. The language of presentation should be as technical as possible for such a problem. Authors should work more with facts and parameters.
Reviewer 2 Report
As attached

Some typos are present.
Line 132 - data is 'sent' through -- not 'sens' through
Line 305 -signal 'bears' a value zero -- not 'bares' a value
Do check the rest.
